# Characteristics and Outcomes of Adult Patients in the PETHEMA Registry with Relapsed or Refractory *FLT3*-ITD Mutation-Positive Acute Myeloid Leukemia

**DOI:** 10.3390/cancers14112817

**Published:** 2022-06-06

**Authors:** David Martínez-Cuadrón, Josefina Serrano, José Mariz, Cristina Gil, Mar Tormo, Pilar Martínez-Sánchez, Eduardo Rodríguez-Arbolí, Raimundo García-Boyero, Carlos Rodríguez-Medina, Carmen Martínez-Chamorro, Marta Polo, Juan Bergua, Eliana Aguiar, María L. Amigo, Pilar Herrera, Juan M. Alonso-Domínguez, Teresa Bernal, Ana Espadana, María J. Sayas, Lorenzo Algarra, María B. Vidriales, Graça Vasconcelos, Susana Vives, Manuel M. Pérez-Encinas, Aurelio López, Víctor Noriega, María García-Fortes, María C. Chillón, Juan I. Rodríguez-Gutiérrez, María J. Calasanz, Jorge Labrador, Juan A. López, Blanca Boluda, Rebeca Rodríguez-Veiga, Joaquín Martínez-López, Eva Barragán, Miguel A. Sanz, Pau Montesinos

**Affiliations:** 1Hospital Universitari i Politècnic La Fe, 46026 Valencia, Spain; boluda_bla@gva.es (B.B.); rodriguez_reb@gva.es (R.R.-V.); barragan_eva@gva.es (E.B.); msanz@uv.es (M.A.S.); montesinos_pau@gva.es (P.M.); 2Hospital Universitario Reina Sofía and Instituto Maimónides de Investigación Biomédica Córdoba (IMIBIC), 14004 Córdoba, Spain; josefina.serrano@iname.com; 3IPO (Istituto Portugues Oncologia), 4200-072 Porto, Portugal; mariz@ipoporto.min-saude.pt; 4Hospital General Universitario de Alicante, 03010 Alicante, Spain; gil_cricor@gva.es; 5Hospital Clínico Universitario, INCLIVA Biomedical Research Institute, 46010 Valencia, Spain; tormo_mar@gva.es; 6Hospital Universitario 12 de Octubre, Complutense University, i+12, CNIO, 28041 Madrid, Spain; mpmartinezsa@yahoo.es (P.M.-S.); jmarti01@med.ucm.es (J.M.-L.); 7Hospital Universitario Virgen del Rocío, 41013 Sevilla, Spain; edurodarb@gmail.com; 8Hospital General Universitari de Castelló, 12004 Castellón, Spain; garcia_rai@gva.es; 9Hospital Universitario de Gran Canaria Doctor Negrín, 35010 Las Palmas, Spain; hematocritico@yahoo.es; 10Hospital Quirón Pozuelo, 28223 Madrid, Spain; carmenmartinezchamorro@hotmail.com; 11Hospital Clínico San Carlos, 28040 Madrid, Spain; marta.polo@salud.madrid.org; 12Hospital San Pedro Alcántara, 10003 Cáceres, Spain; juanmiguel.bergua@salud-juntaex.es; 13Hospital Sao Joao, 4200-319 Porto, Portugal; elianavaleaguiar@gmail.com; 14Hospital General Universitario Morales Meseguer, 30008 Murcia, Spain; mluzamigo@yahoo.com; 15Hospital Universitario Ramón y Cajal, 28034 Madrid, Spain; pilar.herrera@salud.madrid.org; 16Hospital Universitario Fundación Jiménez Díaz, 28040 Madrid, Spain; juan.adominguez@fjd.es; 17Hospital Universitario Central de Asturias, 33011 Asturias, Spain; teresa.bernal@sespa.es; 18Hospital de Coimbra, 3400-091 Coimbra, Portugal; anaespadana@sapo.pt; 19Hospital Universitario Doctor Peset, 46017 Valencia, Spain; sayas_mjo@gva.es; 20Hospital General Universitario de Albacete, 02006 Albacete, Spain; jlalgarra@sescam.jccm.es; 21Department of Hematology, University Hospital of Salamanca (HUS/IBSAL), CIBERONC and Cancer Research Institute of Salamanca-IBMCC (USAL-CSIC), 37007 Salamanca, Spain; mbvidri@usal.es (M.B.V.); mcchillon@saludcastillayleon.es (M.C.C.); 22Hospital de Santa Maria, 1649-028 Lisboa, Portugal; gesteves@medicina.ulisboa.pt; 23ICO-Hospital Germans Trias i Pujol, Badalona, José Carreras Leukemia Research Institute, Universitat Autònoma de Barcelona, 08916 Barcelona, Spain; svives@iconcologia.net; 24Hospital Clínico Universitario de Santiago-CHUS, 15706 Santiago de Compostela, Spain; manuelmateo.perez@usc.es; 25Hospital Arnau de Vilanova, 46015 Valencia, Spain; aureliolopmar@hotmail.com; 26Complejo Hospitalario Universitario A Coruña, 15006 La Coruna, Spain; victor.noriega.concepcion@sergas.es; 27Hospital Universitario Virgen de la Victoria, 29010 Málaga, Spain; mgarciafortes@gmail.com; 28Hospital Universitario de Basurto, 48013 Bilbao, Spain; juanignacio.rodriguezgutierrez@osakidetza.eus; 29Clínica Universidad de Navarra, 31008 Pamplona, Spain; mjcal@unav.es; 30Hospital Universitario de Burgos, 09006 Burgos, Spain; jlabradorg@saludcastillayleon.es; 31Complejo Hospitalario Ciudad de Jaén, 23007 Jaén, Spain; juanantlop@yahoo.es

**Keywords:** acute myeloid leukemia, *FLT3*-ITD mutation, real-world outcomes, relapsed/refractory disease, salvage therapy

## Abstract

**Simple Summary:**

Most adult patients with acute myeloid leukemia (AML) relapse after achieving complete remission with chemotherapy; however, there is no standard second-line (salvage) treatment. We retrospectively investigated 404 patients aged ≥18 years with relapsed/refractory (R/R) AML with an FMS-like tyrosine kinase 3 (*FLT3*) mutation, treated at a PETHEMA (NCT02607059) site between 1998 and 2018. Patients received salvage treatment with intensive therapy (*n =* 261), non-intensive therapy (*n* = 63) or supportive care (*n* = 80). Complete remission was achieved by 48% of patients who received intensive therapy vs. 19% with non-intensive therapy. Intensive/non-intensive therapy prolonged overall survival significantly compared with supportive therapy. Of evaluable patients, 22% received an allogeneic stem-cell transplant after complete remission. The majority of patients with *FLT3*-mutated R/R AML received intensive salvage therapy, with the best outcomes being obtained when intensive salvage treatment was combined with stem-cell transplant.

**Abstract:**

This retrospective study investigated outcomes of 404 patients with relapsed/refractory (R/R) FMS-like tyrosine kinase 3 (FLT3)-internal tandem duplication (ITD) acute myeloid leukemia (AML) enrolled in the PETHEMA registry, pre-approval of tyrosine kinase inhibitors. Most patients (63%) had received first-line intensive therapy with 3 + 7. Subsequently, patients received salvage with intensive therapy (*n* = 261), non-intensive therapy (*n* = 63) or supportive care only (*n* = 80). Active salvage therapy (i.e., intensive or non-intensive therapy) resulted in a complete remission (CR) or CR without hematological recovery (CRi) rate of 42%. More patients achieved a CR/CRi with intensive (48%) compared with non-intensive (19%) salvage therapy (*p* < 0.001). In the overall population, median overall survival (OS) was 5.5 months; 1- and 5-year OS rates were 25% and 7%. OS was significantly (*p* < 0.001) prolonged with intensive or non-intensive salvage therapy compared with supportive therapy, and in those achieving CR/CRi versus no responders. Of 280 evaluable patients, 61 (22%) had an allogeneic stem-cell transplant after they had achieved CR/CRi. In conclusion, in this large cohort study, salvage treatment approaches for patients with *FLT3*-ITD mutated R/R AML were heterogeneous. Median OS was poor with both non-intensive and intensive salvage therapy, with best long-term outcomes obtained in patients who achieved CR/CRi and subsequently underwent allogeneic stem-cell transplant.

## 1. Introduction

The majority of adults with acute myeloid leukemia (AML) show resistance to the first induction chemotherapy or relapse after achieving a first complete remission (CR) [1,2,3]. Prognosis in these patients is dismal, and there is no standard salvage treatment [2,4,5,6]. The goal of salvage therapy is to achieve CR with incomplete peripheral blood count recovery (CRi) or CR in order to perform an allogeneic stem cell transplant (allo-SCT), which appears to be the most curative therapy in this setting [4,5,7]. However, some patients may not receive therapy for relapsed/refractory (R/R) disease, and those who are actively treated may not receive salvage therapy with curative intent [3].

While conventional chemotherapy remains the backbone of salvage therapy, some molecularly targeted agents have been introduced more recently [7,8]. For example, agents targeted at FMS-like tyrosine kinase 3 (*FLT3)* mutations, present in approximately 30% of patients and generally associated with poor outcomes [9], have been approved and are recommended for use in adults with R/R AML [8,10,11]. Second-generation FLT3 inhibitors, such as gilteritinib and quizartinib, have demonstrated superior CR/CRi rates and improved overall survival (OS) compared with standard salvage regimens in patients with R/R AML and an *FLT3* mutation [12,13].

Prognostic factors in R/R AML are not yet well established; for example, the effects on OS of the duration of the first CR/CRi, cytogenetic status, the presence of *FLT3*-internal tandem duplication (ITD) mutations or previous allo-SCT are largely unknown. Reported CR/CRi rates with intensive salvage regimens are approximately 32–66% with cytarabine (Ara-C)- and/or mitoxantrone-based therapies [14,15,16,17,18], up to 55% with FLAG (fludarabine, high-dose Ara-C and granulocyte colony-stimulating factor [G-CSF]) or FLAG-IDA (FLAG plus idarubicin) [19,20,21,22] and 63% with gemtuzumab ozogamicin [23]. Nevertheless, the majority of these studies were performed in relatively small cohorts of patients with varying clinical characteristics, preventing any conclusions regarding the superiority of individual regimens, especially in patients with *FLT3* mutations.

This systematic, retrospective chart review examined real-life outcomes in Spanish and Portuguese patients with R/R *FLT3*-ITD-mutated AML from the Programa Español de Tratamientos en Hematología (PETHEMA) epidemiologic registry.

## 2. Materials and Methods

### 2.1. Study Design

This was a non-interventional, systematic, retrospective chart review of data from patients enrolled in the PETHEMA registry (NCT02607059), which included patients diagnosed with AML, regardless of the treatment administered. The registry was searched for all patients fulfilling the specific inclusion/exclusion criteria of the study. Spanish and Portuguese institutions participated in this study, the protocol of which was approved by the corresponding research ethics board of each institution according to the ethical principles of the Declaration of Helsinki. Informed consent was collected from all patients who were alive at the time of data lock.

### 2.2. Main Inclusion Criteria

Patients diagnosed with de novo or secondary (therapy-related or secondary to myelodysplastic syndrome [MDS] or myeloproliferative syndrome [MPS]) AML and treated at a PETHEMA site between 1 January 1998 and 31 December 2018 were eligible for inclusion in the study (Table 1). R/R disease was defined as failure to achieve CR/CRi (defined as persistence of ≥5% blasts in bone marrow [BM] or peripheral blood [PB], or extramedullary disease) after first-line induction intensive chemotherapy or relapse (defined as reappearance of ≥5% blasts in the BM, PB or an extramedullary site) after first achievement of CR/CRi. In unfit patients treated with non-intensive therapy (defined as hypomethylating agents or low-dose Ara-C [LDAC]-based regimens), disease was classified as resistant only after ≥3–6 cycles, unless the patients showed progression or the physician switched to another line of therapy. *FLT3*-ITD mutations were detected as described in the Appendix A.

### 2.3. Data Extraction

De-identified patient-level data from all patients meeting the inclusion criteria were entered into a secure database, with a data cut-off date of 15 October 2019. Data, entered into an electronic case report form, included patient demographics (age at R/R date (index date), gender); clinical characteristics (date of diagnosis, de novo or secondary AML, prior MDS or MPS, extramedullary involvement, French–American–British (FAB) classification, cytogenetic status at diagnosis according to the Medical Research Council (MRC) criteria [24], mutation status at diagnosis and relapse, Eastern Cooperative Oncology Group (ECOG) performance status and laboratory values at diagnosis and relapse); treatment patterns (first-line treatment regimen (type, start date, end date), stem cell transplant (type, date and number of transplants) and R/R therapy, including any investigational therapies used); and outcomes (response, time, type and location of relapse, development of second neoplasia, date and cause of death and date of last follow up).

### 2.4. Treatment Schedules

Patients were classified in three therapeutic groups according to the intensity of each approach: intensive therapy, non-intensive therapy and supportive care only. Intensive therapy regimens usually included anthracycline plus Ara-C-based regimens, for example, 3 + 7 (idarubicin or daunorubicin and Ara-C), mitoxantrone plus Ara-C, FLAG-IDA, FLAT (fludarabine, Ara-C and topotecan) or ICE (idarubicin, Ara-C and etoposide). The non-intensive therapy group included hypomethylating agents (decitabine or azacitidine at low-doses), FLUGA (fludarabine and Ara-C), FLAG-IDA-Lite (fludarabine, Ara-C, and idarubicin), LDAC or non-intensive regimens in clinical trials. Patients participating in clinical trials of FLT3 inhibitors were included in the intensive or non-intensive therapy groups according to the corresponding planned treatment. Supportive care only group included patients receiving transfusions and other supportive measures, including oral agents to control the white blood cell (WBC) counts (hydroxyurea, melphalan, mercaptopurine or thioguanine).

### 2.5. Endpoints and Outcome Measures

The primary endpoint of the study was OS, defined as time from the start of each salvage therapy to death. Secondary endpoints were morphologic CR, CRi and partial remission (PR; defined according to modified International Working Group criteria [25]), induction death (defined as patients who died after starting salvage therapy but before being assessed), event-free survival (EFS) and frequency of subsequent allo-SCT. PR required all of the hematologic parameters for CR, with ≥50% reduction in blasts to 5–25% [25]. OS data were analyzed according to treatment patterns at R/R episode, with treatment categorized according to intensive, non-intensive, or supportive care. OS data were also analyzed according to the type of first R/R disease episode (defined as primary resistant disease treated with a different induction regimen, primary resistant disease treated with a second identical induction cycle (second induction) or relapse occurring >1 year after first CR or relapse occurring <1 year after first CR).

### 2.6. Statistical Analyses

To address differences in CR and CRi rates among different subsets, comparisons between unrelated variables were performed using χ^2^ and Fisher exact test, as well as the Wilcoxon/Mann–Whitney’s U-test for comparison of continuous variables. Kaplan–Meier estimates were used to calculate unadjusted time-to-event variables and the log-rank test was used to compare them according to the different therapeutic approaches. OS was calculated from the date of AML diagnosis until death in all included patients. EFS was measured from the date of diagnosis until the date of PR/resistant disease, relapse from CR/CRi or death by any cause (whichever occurred first). Multivariate analysis of OS was estimated using a Cox proportional hazards model and included those characteristics with statistical significance in the univariate analysis (*p* < 0.1) as covariates. All *p*-values reported are two-sided. All statistical analyses were performed using the R 2.14.0 software package.

## 3. Results

### 3.1. First-Line Therapy

Overall, 404 patients fulfilling the inclusion criteria were identified. The more frequent first-line active therapy regimens were 3 + 7 (63%) followed by an LDAC-based regimen (usually FLUGA, combining LDAC and oral fludarabine with G-CSF; 10%; Appendix A). Detailed baseline patient demographics and clinical characteristics at initial diagnosis of AML are shown in Table 2.

Among the 403 patients assessed for *FLT3*-ITD status at the initial AML diagnosis, 395 (98%) were *FLT3*-ITD positive, and 8 patients who were *FLT3*-ITD negative initially subsequently tested positive at first R/R episode (one additional patient was not tested at diagnosis, but was *FLT3*-ITD positive at first R/R episode). Patients had an intermediate cytogenetic risk (76%), and mutations in *NPM1* (51%), *IDH1/2* (6%) or *CEBPA* (3%). The median age at diagnosis was 59 years, and 16% had secondary AML. Patients who received non-intensive approaches at first R/R episode were more likely to have the following characteristics at initial diagnosis compared with those receiving intensive salvage regimens: older age (72 years; *p* < 0.0001), more secondary AML (27%; *p* = 0.0007), less extramedullary disease such as hepatomegaly or splenomegaly (15%; *p* = 0.003), lower WBC (26 × 10^9^/L; *p* = 0.02), lower *FLT3*-ITD allelic ratio (0.4; *p* = 0.0004) and more isocitrate dehydrogenase (*IDH*) mutations (14%; *p* = 0.003; Table 2).

Overall, 50% of the 404 patients achieved CR/CRi after the first induction cycle and 10% achieved CR/CRi after 2–3 cycles, while 9% had PR and 31% failed to respond. Consequently, 50% of patients were considered to have primary refractory disease (including those not achieving CR/CRi after the first induction cycle) and 50% had relapsed disease after first CR/CRi. Fifteen percent of patients had an early relapse (CR1 duration < 12 months), 36% had a late relapse (CR1 duration > 12 months), 41% had primary resistance (after only 1 cycle) and 8% had primary resistance (after only 1 cycle), but were salvaged with a second identical 3 + 7 cycle (second induction).

### 3.2. Salvage Therapy

Details of salvage therapy for the first R/R episode are provided in Appendix A. The mean time from initial AML diagnosis to subsequent R/R disease was 8.9 months (range, 1.6–127.3 months) in the overall study population. The mean time from initial diagnosis to R/R disease was shorter in patients who went on to receive supportive therapy only (8.1 months; range 1.6–79.5 months) than in those who received non-intensive (9.7 months; range, 1.9–104.5 months) or intensive salvage therapy (8.9 months; range, 1.8–127.3 months); however, between-group differences were not significant (Kruskal–Wallis χ^2^ 2.9595; degrees of freedom 2; *p* = 0.2277). Up to 3% of first R/R episodes had extramedullary involvement (with or without concurrent bone marrow infiltration).

In the overall study population, 80 (20%) received supportive care only and 324 (80%) received an active salvage treatment (Figure 1).

The majority of patients receiving active salvage therapy were treated with FLAG-based regimens (32%), 3 + 7 (13%) or other intensive regimens (16%). Detailed information regarding the salvage treatment received is given in Appendix A.

Patient characteristics at the first R/R episode in the overall study population and according to the salvage therapy received are shown in Table 3.

Patients receiving supportive therapy only were significantly older at the first R/R episode and had significantly poorer performance status than those who received intensive or non-intensive salvage therapy (both *p* < 0.001; Table 3). Additionally, significantly more patients treated in the non-intensive therapy group received salvage therapy through clinical trial participation than in the intensive therapy group (*p* < 0.001; Table 3).

Among the 324 patients who received active salvage therapy, response to treatment data were available in 280 patients (86%; Table 4). The CR/CRi rate was 42% (119/280) and 134 patients (48%) had resistant disease (including PR).

More patients achieved CR/CRi with intensive salvage therapy (45% CR and 3% CRi) compared to those who received non-intensive salvage therapy (15% CR and 4% CRi; *p* < 0.001).

### 3.3. Stem Cell Transplantation in Second-Line Therapy

Overall, 67 patients received allo-SCT during second-line treatment; six (2%) of these were performed as part of the first salvage regimen in patients with active AML. Three patients (1%) received an autologous-SCT after achieving CR/CRi. All patients who underwent allo-SCT were induced with an intensive salvage regimen for the first R/R episode, so an allo-SCT in CR/CRi was performed in 61 out of the 223 evaluable patients in the intensive therapy group (27%).

### 3.4. Survival Analysis

Data for 401 patients were evaluable for the OS analyses. At data cut-off, the median follow-up was 5 (range, 0–210) months in surviving patients. A total of 27 patients (10%) died during salvage treatment before the first treatment assessment. The median OS of the entire cohort was 5.55 months (95% confidence interval [CI] 4.2, 6.7; Table 4 and Appendix A).

The univariate subgroup analysis showed that MRC cytogenetic risk score did not have a significant effect on OS (*p* = 0.652; Figure 2a), but *IDH* mutation status (*p* = 0.02; Figure 2b), type of R/R episode (*p* = 0.002; Figure 2c) and age at relapse (*p* < 0.001) did. No differences in OS were observed between patients with de novo AML and secondary AML (*p* = 0.065; Figure 2d), between patients diagnosed with AML secondary to MDS/MPS or not (*p* = 0.230) or between patients with de novo AML and those with AML secondary to MDS/MPS (*p* = 0.201).

Median OS was significantly prolonged in patients receiving intensive or non-intensive salvage therapy compared to those receiving supportive therapy only (7.2 and 6.2 months, respectively, vs. 1.0 month; *p* < 0.001; Table 4 and Figure 3a).

Response to first salvage treatment significantly impacted OS, with achievement of CR/CRi being associated with significantly improved OS compared with PR or resistance (13.6 vs. 6.3 or 4.9 months; *p* < 0.001; Figure 3b). Additionally, when response to salvage treatment at first R/R was categorized according to achievement of CR/CRi versus no CR/CRi, OS was significantly improved in those with CR/CRi (13.6 vs. 3.9 months; *p* < 0.001; Figure 3c). However, in patients achieving CR/CRi after salvage therapy, there was no significant difference when assessed by the type of post-remission therapy (i.e., auto- or allo-SCT vs. no SCT; 10.3 vs. 15.3 vs. 14.5 months; *p* = 0.2); however, few long-term survivors were mainly among the SCT group (Figure 3d).

Among the 280 patients with EFS data, median EFS was 0.03 months (95% CI 0.03, 0.03). Furthermore, at 1 and 2 years, 17% and 11% of patients had EFS, respectively (Table 4).

#### Multivariate Analysis of Overall Survival

In a multivariate Cox regression analysis, lack of *IDH* mutation and receiving a second identical induction cycle had a positive impact on OS, with hazard ratios (HR) of 0.74 (95% CI 0.55, 0.99; *p* = 0.04) and 0.45 (95% CI 0.26, 0.79; *p* = 0.005), respectively. In contrast, not receiving active salvage treatment (HR 5.75; 95% CI 4.02, 8.26; *p* < 0.001) and being aged 60–70 years (HR 1.54; 95% CI 1.05, 2.25; *p* = 0.03) were independent prognostic factors for reduced OS.

## 4. Discussion

This study provides real-life evidence on characteristics, treatment patterns and outcomes of patients with R/R *FLT3*-ITD mutation-positive AML in the pre-FLT3 inhibitors era. In this large cohort of patients enrolled in the PETHEMA registry, salvage treatment approaches were heterogeneous, from supportive care only to non-intensive or intensive therapy regimens, followed by allo-SCT. Median OS and EFS were poor for both non-intensively and intensively treated patients, with the best long-term outcomes obtained in patients achieving CR/CRi and subsequently receiving SCT.

To our knowledge, this is the largest real-world study focusing specifically on outcomes in R/R patients with *FLT3*-ITD mutation-positive AML. The French DATAML registry study by the Toulouse–Bordeaux group included 160 patients with *FLT3*-ITD mutation-positive R/R AML [26], while other studies were performed in smaller patient populations [27,28]. In the French study, 294 *FLT3*-ITD-mutated patients who received FLT3 inhibitors as an intensive first-line treatment regimen were followed-up, resulting in 160 R/R episodes (14 of them treated with second generation FLT3 inhibitors and not evaluated for efficacy). The characteristics at initial diagnosis of this cohort was very similar to that observed in this study (median age of 57 years, secondary AML in 11%, extramedullary disease in 41%, *IDH* mutations in 11%, median WBC of 51 × 10^9^/L and *NPM1* co-mutations in 65%). It should be noted that this study included R/R patients who received first-line non-intensive regimens (~14%), providing a broader picture of routine clinical practice. This study also showed that patients who received non-intensive salvage therapy regimens (LDAC-based or hypomethylating agents) had less proliferating *FLT3*-ITD-mutated AML with more frequent features common to leukemia seen in older patients (e.g., secondary AML, *IDH* mutations) [29,30,31]. Because the current study also included these unfit patients, it is likely that the patterns of care for salvage therapy were slightly different to those reported by the French group: most patients in both studies received intensive salvage (65% and 67%, respectively) or supportive care only (20% and 27%); however, more non-intensive therapy regimens were used in this study (16% vs. 6%). In a real-world study of treatment practices and clinical outcomes among Australian patients (*n* = 73) with newly diagnosed *FLT3* mutation-positive AML in the pre-midostaurin era, the study population was broadly similar to that of our study in terms of adverse cytogenetic risk (7%) and *NPM1* co-mutation positivity (60%) [27]. However, no outcomes regarding rates of R/R disease, salvage therapies received or outcomes after salvage therapy were reported in the previous study [27]. Another real-world study of 284 US-based patients with AML evaluated patient characteristics, treatment patterns, as well as outcomes after diagnosis of R/R disease [28]. However, only 42 of the 284 patients in this study had *FLT3* mutations, and no outcomes specific to those patients were given [28].

The current analysis showed a relatively high CR/CRi rate (42%) after salvage therapy for first R/R episode, with an expected lower rate (19%) after non-intensive therapy and a higher rate (48%) after intensive therapy. Interestingly, the French study showed a CR/CRi rate of 49% after intensive salvage therapy, with a subsequent allo-SCT rate of 29%, which is comparable to our data (27%) [26]. The median OS in our series was 5.5 months overall, 1.0 month with supportive care only, 6.2 months with non-intensive therapy and 7.2 months with intensive therapy (similar to 7.5 months reported DATAML cohort after intensive treatment [26]). Although median OS in this study was not significantly different between the intensive and non-intensive therapy approaches, only a few long-term survivors achieved CR/CRi and underwent SCT after intensive salvage therapy. Interestingly, patients with an *IDH1* mutation had worse prognosis, as previously described by Wattad and colleagues [32] in a large study on primary refractory AML including patients with *FLT3*-mutated and wild-type disease, while those with *IDH2*-mutated AML had better outcomes. We also found that late relapses, as well as CR/CRi after a second identical induction cycle, was associated with better OS than early relapses or primary resistant disease, as previously reported for patients with R/R *FLT3*-mutated and wild-type AML [26].

Quality of life (QoL) and patient-reported outcomes (PROs) are increasingly valuable endpoints in studies of hematologic malignancies, with the relevance and value of novel preference-based measures in myeloid malignancies, such as AML, being particularly important [33]. Nevertheless, the retrospective nature of our study did not allow for the collection of this information. Future studies of QoL and PROs in patients with AML are needed.

Real-life studies focusing on R/R *FLT3*-ITD-mutated AML patients could be useful to estimate the potential gains from switching salvage strategies from chemotherapy regimens to second generation FLT3 inhibitors. Patient and disease characteristics in our study were relatively similar to those of the QuANTUM-R and ADMIRAL studies evaluating gilteritinib and quizartinib [12,13] versus standard of care. Of note, median OS in the control arms of QuANTUM-R and ADMIRAL were 4.7 and 5.6 months, respectively, which is slightly lower than that reported in our overall study population (including the supportive care only group). We should also highlight that the CR/CRi and SCT rates in this study and in the French registry [26] were higher than in the control arms of both phase 3 studies [12,13]. Nevertheless, we cannot compare real-life outcomes with that observed in clinical trials. For example, QuANTUM-R included patients with worse characteristics (i.e., those with late relapse were excluded), probably leading to worse outcomes in the experimental and standard-of-care arms. Real-world studies describing conventional care could be useful for selection of optimal salvage therapy in patients with *FLT3*-ITD-mutated AML, even in the era of targeted therapies. While outcomes in *FLT3* mutation-positive patients have improved with FLT3-targeted agents compared with standard salvage therapy [12,13], these new therapies may not be available in many countries.

## 5. Conclusions

In conclusion, this is the largest real-world study in patients with R/R *FLT3* mutation-positive AML in the pre-FLT3-inhibitors era. Thus, these data are important to help inform the design of future clinical trials in this setting. A sizable proportion of patients received supportive care only or a non-curative approach. Given the acceptable safety profile of new targeted agents, we believe that unfit R/R *FLT3*-ITD-mutated AML patients could benefit from these new drugs (e.g., gilteritinib or quizartinib). The majority of patients with R/R *FLT3*-ITD-mutated AML received intensive therapy, with the best outcomes obtained after achieving CR/CRi, especially when followed by allo-SCT. Beyond the results of the randomized QuANTUM-R and ADMIRAL studies, the true benefit of second-generation FLT3 inhibitors for fit patients with AML needs to be confirmed in real-life studies.

## Figures and Tables

**Figure 1 cancers-14-02817-f001:**
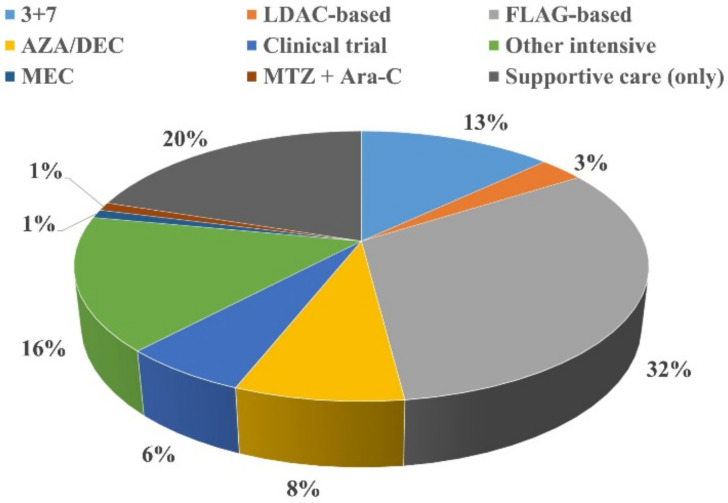
Salvage therapies received by the study population (*n* = 404). Overall, 3 + 7, 3 days of daunorubicin plus 7 days of cytarabine. AraC: cytarabine; AZA/DEC: azacitidine or decitabine; FLAG: fludarabine, high-dose cytarabine and granulocyte colony-stimulating factor; LDAC: low-dose cytarabine; MEC: mitoxantrone, etoposide and cytarabine; MTZ: mitoxantrone.

**Figure 2 cancers-14-02817-f002:**
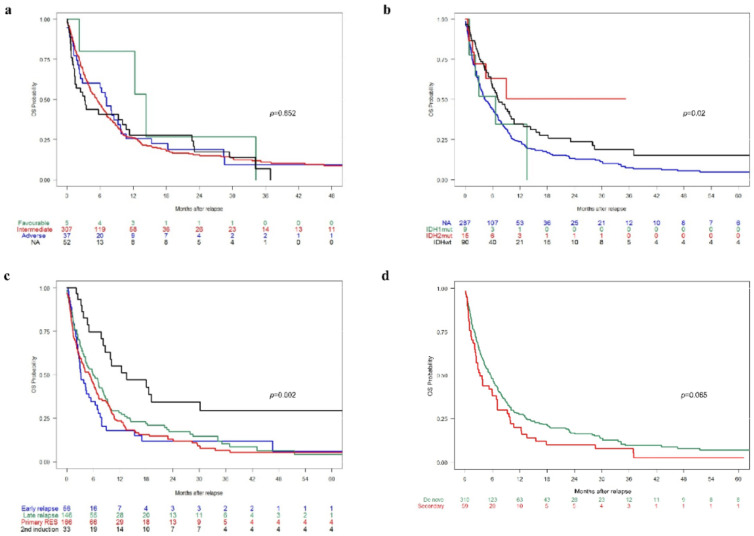
Kaplan–Meier plots for overall survival (OS) according to (**a**) Medical Research Council risk; (**b**) isocitrate dehydrogenase (IDH) mutation status; (**c**) the type of relapsed/refractory disease; and (**d**) type of acute myeloid leukemia. mut: mutation; NA: not available; RES: resistance; wt: wild type.

**Figure 3 cancers-14-02817-f003:**
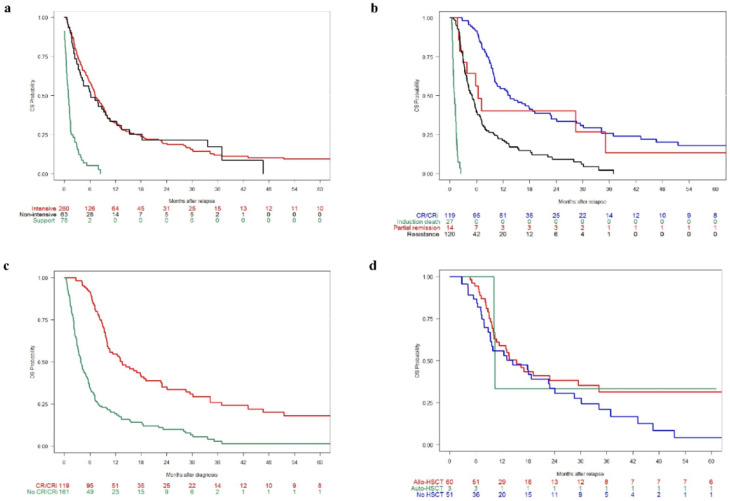
Overall survival (OS) in treated patients according to (**a**) therapeutic approach at first relapsed/refractory episode; (**b**) response to salvage treatment; (**c**) complete remission (CR)/complete remission with incomplete peripheral blood count recovery (CRi) vs. no CR/CRi after first salvage treatment; and (**d**) in patients achieving CR/CRi after salvage therapy, according to post-remission therapy (patients treated with direct allogeneic stem-cell transplantation (allo-SCT) were excluded [*n* = 114]). PR: partial remission.

**Table 1 cancers-14-02817-t001:** Main inclusion and exclusion criteria.

Inclusion Criteria	Exclusion Criteria
Aged ≥18 years when diagnosed with AML Relapsed or refractory AML Tested positive for an *FLT3*-ITD mutation during the study period using a locally available test Received active therapy ^†^ in first line	Acute promyelocytic leukemia (M3 AML) or mixed phenotype AML

AML: acute myeloid leukemia; FLT3: FMS-like tyrosine kinase 3; ITD: internal tandem duplication. ^†^ Oral hydroxyurea, mercaptopurine, thioguanine or melphalan, alone or in combination, were considered as supportive care only (not active therapy).

**Table 2 cancers-14-02817-t002:** Baseline patient demographics and clinical characteristics of the study population at initial acute myeloid leukemia diagnosis according to subsequent salvage treatment received.

Characteristic at Initial Diagnosis	Total (*N* = 404)		Intensive (*N* = 261)		Non-Intensive (*N* = 63)		Supportive Care Only (*N* = 80)		*p*-Value
	Median (Range)	*n* (%)	Median (Range)	*n* (%)	Median (Range)	*n* (%)	Median (Range)	*n* (%)	
Age (years)	59 (18–86)	404 (100)	52 (18–76)	261 (100)	72 (33–83)	63 (100)	68 (37–86)	80 (100)	<0.001 *
<60		208 (51)		180 (69)		12 (19)		16 (20)	<0.001
≥60		196 (49)		81 (31)		51 (81)		64 (80)	
Sex		402 (100)		261 (100)		61 (100)		80 (100)	
Male		196 (49)		126 (48)		34 (56)		36 (45)	0.4
Female		206 (51)		135 (52)		27 (44)		44 (55)	
Type of AML		372 (100)		238 (100)		62 (100)		72 (100)	
De novo		313 (84)		210 (88)		45 (73)		58 (81)	0.007
Secondary		59 (16)		28 (12)		17 (27)		14 (19)	
Therapy-related AML		367 (100)		234 (100)		62 (100)		71 (100)	
Yes		26 (7)		13 (6)		8 (13)		5 (7)	
No		341 (93)		221 (94)		54 (87)		66 (93)	0.13
Secondary to MDS/MPS		367 (100)		234 (100)		62 (100)		71 (100)	
Yes		28 (8)		11 (5)		9 (15)		8 (11)	0.02
No		339 (92)		223 (95)		53 (85)		63 (89)	
FAB subtype		404 (100)		261 (100)		63 (100)		80 (100)	
M0		23 (6)		17 (7)		4 (6)		2 (2)	0.16
M1		69 (17)		43 (16)		14 (22)		12 (15)	
M2		46 (11)		27 (10)		6 (10)		13 (16)	
M4		107 (26)		68 (26)		14 (22)		25 (31)	
M5		65 (16)		51 (20)		9 (14)		5 (6)	
M6		5 (1)		2 (1)		2 (3)		1 (1)	
M7		2 (1)		1 (0)		1 (2)		0 (0)	
NA		87 (22)		52 (20)		13 (21)		22 (27)	
Extramedullary disease		285 (100)		174 (100)		55 (100)		56 (100)	
Yes		75 (26)		58 (33)		8 (15)		9 (16)	
No		210 (74)		116 (67)		47 (85)		47 (84)	0.003
WBC, ×10^9^/L	51.4 (0.6–365.5)	379 (100)	51.2 (0.9–65.5)	247 (100)	26 (0.6–384.4)	59 (100)	67.5 (0.9–292.3)	73 (100)	0.02*
≤10		70 (18)		45 (18)		15 (25)		10 (14)	0.22
>10		309 (82)		202 (82)		44 (75)		63 (86)	
Cytogenetics		404 (100)		261 (100)		63 (100)		80 (100)	
Normal		253 (630)		168 (64)		42 (67)		43 (54)	0.31
Abnormal		95 (24)		63 (24)		13 (21)		19 (24)	
No metaphases		33 (8)		16 (6)		6 (10)		11 (14)	
NA		23 (6)		14 (5)		2 (3)		7 (9)	
MRC cytogenetic risk		404 (100)		261 (100)		63 (100)		80 (100)	
Favorable		5 (1)		3 (1)		2 (3)		0 (0)	0.23
Intermediate		309 (76)		204 (78)		51 (81)		54 (67)	
Adverse		38 (9)		24 (9)		4 (6)		10 (13)	
NA		52 (13)		30 (12)		6 (10)		16 (20)	
*FLT3*-ITD mutation		403 (100) ^†^		261 (100)		63 (100)		79 (100) ^†^	
Positive		395 (98)		259 (99)		59 (94)		77 (97)	0.02
Negative		8 (2) ^#^		2 (1) ^#^		4 (6) ^#^		2 (3) ^#^	
*FLT3*-ITD allelic ratio	0.65 (0–10.55)	281 (100)	0.7 (0–10.55)	166 (100)	0.4 (0–7.4)	57 (100)	0.58 (0–4.7)	58 (100)	
<0.06		23 (8)		4 (2)		11 (19)		8 (14)	0.004 *
≥0.06–0.5		93 (33)		53 (32)		22 (39)		18 (31)	
≥0.5–0.8		63 (22)		39 (23)		10 (18)		14 (24)	<0.001
≥0.8		102 (36)		70 (42)		14 (25)		18 (31)	
*NPM1* mutation status		404 (100)		261 (100)		63 (100)		80 (100)	
Positive		205 (51)		127 (49)		39 (62)		39 (49)	0.18
Negative		155 (38)		100 (38)		20 (32)		35 (44)	
NA		44 (11)		34 (13)		4 (6)		6 (7)	
*CEBPA* mutation status		404 (100)		261 (100)		55 (100)		80 (100)	
Positive		11 (3)		9 (3)		1(2)		1 (1)	0.32
Negative		78 (19)		44 (17)		13 (24)		21 (26)	
NA		315 (78)		208 (80)		41 (75)		58 (72)	
*IDH* mutation status		404 (100)		261 (100)		63 (100)		80 (100)	
*IDH1* positive		9 (2)		6 (2)		2 (3)		1 (1)	
*IDH2* positive		15 (4)		6 (2)		7 (11)		2 (2)	
Negative		92 (23)		57 (22)		21 (33)		14 (17)	
NA		288 (71)		192 (74)		33 (52)		63 (79)	0.003

AML: acute myeloid leukemia; BM: bone marrow; FAB: French–American–British; IDH: isocitrate dehydrogenase; ITD: internal tandem duplication; MDS: myelodysplastic syndrome; MPS: myeloproliferative syndrome; MRC: Medical Research Council; NA: not available; NPM1: nucleophosmin 1 gene; PB: peripheral blood; R/R: relapsed/refractory; WBC: white blood cells. * *p*-values compare continuous variables. ^†^ 1 patient had no data regarding *FLT3*-ITD mutation status at diagnosis. ^#^ 8 patients were *FLT3*-ITD-mutation negative at diagnosis but positive after R/R episode, and 1 patient tested only at R/R episode was positive.

**Table 3 cancers-14-02817-t003:** Demographic and baseline characteristics of the study population at first relapsed/refractory episode according to salvage therapy.

Characteristic	Total (*N* = 404)		Intensive (*N* = 261)		Non-Intensive (*N* = 63)		Supportive Care Only (*N* = 80)		*p*-Value
	Median (Range)	*n* (%)	Median (Range)	*n* (%)	Median (Range)	*n* (%)	Median (Range)	*n* (%)	
Time to R/R, ^†^ (months)	4.99 (1.58–79.48)		4.25 (1.81–127.3)		5.91 (1.87–104.5)		5.44 (1.58–79.48)		0.23
Age (years)	60 (18–86)	401 (100)	52 (18–77)	259 (100)	73 (34–84)	63 (100)	69 (37–86)	77 (100)	
<60		190 (47)		167 (64)		10 (16)		13 (17)	<0.001 *
≥60		211 (53)		93 (36)		53 (84)		65 (83)	<0.001
ECOG PS	1 (0–4)	126 (100)	1 (0–4)	79 (100)	1 (0–2)	26 (100)	1 (0–4)	21 (100)	0.001 *
0		43 (34)		31 (39)		10 (38)		2 (10)	0.02
1		58 (46)		36 (46)		13 (50)		9 (43)	
2		16 (13)		8 (10)		3 (12)		5 (24)	
3		6 (5)		3 (4)		0 (0)		3 (14)	
4		3 (2)		1 (1)		0 (0)		2 (10)	
WBC, ×10^9^/L	5.3 (0.2–283)	114 (100)	6.5 (0.4–269)	67 (100)	3.4 (0.6–108)	26 (100)	5.4 (0.2–282)	21 (100)	0.06 *
≤10		71 (62)		40 (60)		18 (69)		13 (62)	0.7
>10		43 (38)		27 (40)		8 (31)		8 (38)	
Hemoglobin, g/dL	10 (4.6–16.1)	113 (100)	9.7 (4.6–16.1)	67 (100)	10.8 (7.8–15.3)	25 (100)	9.6 (6.6–13.5)	21 (100)	0.14 *
≤10		59 (52)		38 (57)		9 (36)		12 (57)	0.18
>10		54 (48)		29 (43)		26 (64)		9 (43)	
Platelet count, ×10^9^/L	81 (1.5–984)	113 (100)	82 (1.5–984)	67 (100)	101 (10–467)	25 (100)	61 (12–294)	21 (100)	0.25 *
≤50		44 (39)		26 (39)		8 (32)		10 (48)	
>50		69 (61)		41 (61)		17 (68)		11 (52)	0.56
PB blasts, %	15 (0–100)	115 (100)	17 (0–100)	68 (100)	11 (0–96)	26 (100)	17 (0–100)	21 (100)	0.70 *
≤50		81 (70)		44 (65)		21 (81)		16 (76)	0.25
>50		34 (30)		24 (35)		5 (19)		5 (24)	
BM blasts, %	45 (0–100)	121 (100)	40 (0–100)	78 (100)	45 (5–99)	25 (100)	46 (9–100)	18 (100)	0.32 *
≤50		68 (56)		44 (56)		14 (56)		10 (56)	0.99
>50		53 (44)		34 (44)		11 (44)		8 (44)	
Previous SCT		403 (100)		261 (100)		63 (100)		79 (100)	
No		318 (79)		195 (75)		56 (89)		67 (85)	0.08
Autologous		33 (8)		26 (10)		2 (3)		5 (6)	
Allogeneic		52 (13)		40 (15)		5 (8)		7 (9)	
Clinical trial salvage		404 (100)		261 (100)		63 (100)		-	<0.001
Yes		24 (6)		4 (2)		20 (32)		-	
No		380 (94)		257 (98)		43 (68)		-	

BM: bone marrow; ECOG PS: Eastern Cooperative Oncology Group performance status; PB: peripheral blood; R/R: relapsed/refractory; SCT: stem cell transplantation; WBC: white blood cells. * *p*-values compare continuous variables. ^†^Median time from induction to R/R disease.

**Table 4 cancers-14-02817-t004:** Outcomes according to second line of treatment.

Variable	All Patients (*N* = 404)	Intensive (*N* = 261)	Non-Intensive (*N* = 63)	Supportive (*N* = 80)	*p*-Value *
Response, *n* (%)	*n* = 280	*n* = 223	*n* = 57	-	
ORR (CR + CRi)	119 (42)	108 (48)	11 (19)	-	<0.001
CR	110 (39)	101 (45)	9 (15)	-	
CRi	9 (3)	7 (3)	2 (4)	-	
PR	14 (5)	12 (5)	2 (4)	-	
Resistance	120 (43)	82 (37)	38 (67)	-	
Induction death	27 (10)	21 (9)	6 (11)	-	
OS, months	*n =* 401	*n* = 260	*n* = 63	*n* = 78	
Median (95% CI)	5.5 (4.2–6.7)	7.2 (6.6–9.3)	6.2 (4.2–10.7)	1.0 (0.6–1.2)	<0.001
OS, %					
At 1 year (95% CI)	25 (20–30)	34 (28–42)	33 (23–49)	-	
At 2 years (95% CI)	16 (11–20)	20 (15–27)	22 (12–39)	-	
At 5 years (95% CI)	7 (2–4)	9 (6–16)	-	-	
EFS, months	*n* = 280	*n* = 223	*n* = 57	-	
Median (95% CI)	0.03 (0.03–0.03)	0.03 (0.03–1.6)	0.03 (0.03–0.03)	-	0.008
EFS, %					
At 1 year (95% CI)	17(13–22)	20 (15–26)	10 (5–22)	-	
At 2 years (95% CI)	11 (7–15)	14 (9–20)	8 (3–20)	-	
At 5 years (95% CI)	7 (3–11)	8 (4–14)	-	-	

CI: confidence interval; CR: complete remission; CRi: complete remission with incomplete peripheral blood count recovery; EFS: event-free survival; ORR: overall response rate; OS: overall survival; PR: partial remission. * *p*-values are for comparisons between therapeutic approaches.

## Data Availability

Data are the property of the PETHEMA foundation. Data may be available from the corresponding author upon reasonable request.

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
