# Peer review of "Characteristics and Outcomes of Adult Patients in the PETHEMA Registry with Relapsed or Refractory FLT3-ITD Mutation-Positive Acute Myeloid Leukemia"

_cancers, 2022, doi:10.3390/cancers14112817_

Round 1

Reviewer 1 Report

This study investigates outcome in a retrospective cohort of 404 patients with relaped/refractory FLT3-internal tandem duplication acute myeloid leukemia enrolled in the PETHEMA registry, before approval of tyrosine kinase inhibitors targeting FTL3. More patients obtained a CR/Cri with intensive salvage therapy (48%) compared to non-intensive salvage regimens (19%). Overall survival was significantly prolonged in patients achieving CR/Cri versus non responders and in those receiving anti-leukemic therapy versus those receiving supportive treatment.

MAJOR ISSUES

1. Quality of Life (QoL) and patient Reported Outcomes (PROs) are increasingly important endpoints in studies of hematological malignancies. Were data on Quality of Life available in the PETHEMA registry for the patients included in this study? If yes, data on QoL and PROs should be included and discussed in the manuscript. If this information was not collected and/or is not fully available in the PETHEMA registry, the authors should then include a paragraph in the discussion acknowledging the lack of this information and quoting the relevance of future studies on these important aspects.

2. When discussing the need for future studies on QoL and PROs, the relevance and value of novel preference-based measures in myeloid malignancies should be quoted, also referring to recent reports applied to myelodysplastic syndromes (Gamper EM, et al. The EORTC QLU-C10D was more efficient in detecting clinical known group differences in myelodysplastic syndromes than the EQ-5D-3L. J Clin Epidemiol. 2021;137:31-44).

3. Were there differences in the investigated outcomes based on the distinction between de novo cases versus cases evolved from myelodysplastic syndromes or myeloproliferative neoplasms?

MINOR ISSUES

4. The manuscript may benefit from a certain reduction in length (10% approximately)

Author Response

#Reviewer 1

Comments and Suggestions for Authors

This study investigates outcome in a retrospective cohort of 404 patients with relapsed/refractory FLT3-internal tandem duplication acute myeloid leukemia enrolled in the PETHEMA registry, before approval of tyrosine kinase inhibitors targeting FTL3. More patients obtained a CR/Cri with intensive salvage therapy (48%) compared to non-intensive salvage regimens (19%). Overall survival was significantly prolonged in patients achieving CR/Cri versus non responders and in those receiving anti-leukemic therapy versus those receiving supportive treatment.

Major Issues

  1. Quality of Life (QoL) and patient Reported Outcomes (PROs) are increasingly important endpoints in studies of hematological malignancies. Were data on Quality of Life available in the PETHEMA registry for the patients included in this study? If yes, data on QoL and PROs should be included and discussed in the manuscript. If this information was not collected and/or is not fully available in the PETHEMA registry, the authors should then include a paragraph in the discussion acknowledging the lack of this information and quoting the relevance of future studies on these important aspects.

Authors’ response: We thank the reviewer for this suggestion. However, as this retrospective study included patient data starting from 1998, no data were collected regarding quality of life or patient-reported outcomes. We have included a new paragraph in the discussion (page 13), which highlights the need for future studies of QoL and PROs in patients with AML, as the reviewer has suggested.

  1. When discussing the need for future studies on QoL and PROs, the relevance and value of novel preference-based measures in myeloid malignancies should be quoted, also referring to recent reports applied to myelodysplastic syndromes (Gamper EM, et al. The EORTC QLU-C10D was more efficient in detecting clinical known group differences in myelodysplastic syndromes than the EQ-5D-3L. J Clin Epidemiol. 2021;137:31-44).

Authors’ response: As stated in our response to this reviewer’s previous comment, no data on QoL or PROs were collected in our study; however, we have added a paragraph to the Discussion (page 13) that highlights the need for future studies to assess QoL and PROs in patients with AML.

  1. Were there differences in the investigated outcomes based on the distinction between de novo cases versus cases evolved from myelodysplastic syndromes or myeloproliferative neoplasms?

Authors’ response: No differences in outcomes were observed between de novo cases versus those evolved from MDS or myeloproliferative syndrome. We have added a new sentence describing this to the Results (Section 3.4, page 9) and a new panel to Figure 2 (d) showing the OS probability for de novo versus secondary AML.

Minor Issues

  1. The manuscript may benefit from a certain reduction in length (10% approximately).

Authors’ response: Following the reviewer´s request, we have edited the manuscript text to reduce the word count. The overall length of the manuscript is now 3395 words. Given that the journal’s instructions for authors recommends that original research articles are a minimum of 3000 words and that our manuscript now includes an additional multivariate analysis of overall survival, we feel that this length is appropriate.

Reviewer 2 Report

The authors presented an original manuscript focusing on real-world data of adults FLT3-ITD positive R/R AML. The paper is of interest, considering the huge cohort of patients analyzed, and the potential implications in subsequent studies, including targeted-therapy data. Regardless, the study-cohort is inhomogeneous concerning age, leukemic genomics, inclusion of therapy-related AMLs, and first and second-line therapeutic approaches. Second, in the response-to-therapy assessment, the authors did not include any MRD data, importantly impacting on AML outcomes. Additionally, the study design and results are complex and do not immediately impact on the reader as presented in the paper, especially regarding first-line and second-line treatment effects on outcomes.

Major revisions:

  1. Simple summary and Abstract do not adequately present the main manuscript and requires extensive revisions:

- simple summary, lines 51-52 contrast with discussion, regarding the efficacy of FLT3-inhibitor therapy. Additionally, no data on FLT3-inhibitors are reported in the manuscript results section.

-simple summary, line 53: the manuscript focuses on FLT3 positive R/R AMLs: please edit the text

- simple summary, lines 54-59: please clarify whether salvage therapy refers to second-line treatment only and does not include first-line treatment data.

- simple summary, lines 59-60: the main manuscript does not support this conclusion

- the abstract should include Introduction, Material and Methods, and Conclusions section/information, beyond the results. Please integrate it

-abstract, line 65: please specify that active salvage therapy includes only intensive and non-intensive therapy.

-abstract, lines 69-72: please clarify whether OS data refer to first-line or second-line therapy.

2) Materials and Methods section:

- inclusion and exclusion criteria are complex and reported unclearly. A Table summarizing them may improve the manuscript.

- line 121: the authors should clarify why they included secondary AMLs together with de novo AMLs, considering differences in outcomes among them. Survival analysis could be performed separately for these two groups.

- lines 131-132: the authors should detail in the supplementary section the adopted methods for FLT3-ITD detection, especially considering that the analysis was performed by different laboratories without centralization to a referral lab.

- line 132: the authors should explain why they included patients without FLT3-ITD assessment at first diagnosis, as well as those patients showing FLT3-ITD only at relapse (Table 1). This could affect the study results.

- line 133: please define active first-line therapy. Considering therapy data complexity, the authors should name first-line (front-line) and second-line (salvage) therapy uniformly all over the manuscript.

-line 135: please clarify why second-generation FLT3-inhibitors were excluded and whether first-generation FLT3-inhibitor were included in the study

-lines 138-150: it would be useful to include response to therapy evaluation by MFC-MRD, to better understanding subsequent outcomes data.

- lines 142-146: the authors should clarify why they did not classify AMLs according to the 2016 Revision of WHO classification.

-line 144: please specify which genetic aberrations are included in each cytogenetic category

-lines 152-165: please add some information on SCT (first-line or second line treatment, or both?).

- line168: please define partial remission

- line 169: please clarify whether induction death includes death for any cause or only disease-related death

5) Results:

- line 228: it would be interesting to have data on the time interval between first- and R/R diagnosis

- lines 267-268: please report the reasons for missing follow-up data in 44 patients

-lines 267-277: please move to survival data section

- survival analysis: it would be interesting to perform a multivariable analysis to show what effectively impacts on outcomes, considering the complexity of the study-cohort.

4) Table 1:

-  the age range 17-86 contrasts with the inclusion criterium age ≥ 18 years. Please clarify this discrepancy

- it would be interesting to analyze young adults (18-25 years old) as a separate category, considering the biological differences of AMLs in this age group compared to older patients.

- the authors may add a raw reporting front-line therapy data, including any SCT, if performed. The same could be done in Table 2

5) Figure 2c:  the 2nd Induction category shown in figure 2c is not cited in the manuscript. Please clarify it.

6) Figure 2b: Induction death category should be removed from the analysis, being not comparable with the other three categories and potentially hampering OS results.

7) Supplementary: it would be interesting to include a table matching first-line and second line treatments

Minor revisions:

1) Simple summary:

- line 55: 48% cannot be reported in digits at the beginning of a sentence. Please edit it.

2) Materials and Methods:

-line 121: please define secondary AML, and then remove its definition from the Results section (line 212)

- lines 124, 126, 128, 141, 142, 147, 149, 162, 173: please substitute the i.e. with a complete list of all the definitions/variables pertinent to each specific brackets.

-line 160: please detail LDAC therapy

- lines 163-164: please remove Supplementary Table 1 referral from the Material and Methods section, because it belongs to the Results section.

3) Results:

- line 217: please change allelic ratio to FLT3-ITD allelic ratio

-lines 278-285: the number of patients undergoing SCT after CR/CRi is reported twice. Please edit it.

4) Table 1: age < 60 years, total, %: please change 501 to 51

5) Figures 2 and 3: please add p values

Author Response

#Reviewer 2

Comments and Suggestions for Authors

The authors presented an original manuscript focusing on real-world data of adults FLT3-ITD positive R/R AML. The paper is of interest, considering the huge cohort of patients analyzed, and the potential implications in subsequent studies, including targeted-therapy data. Regardless, the study-cohort is inhomogeneous concerning age, leukemic genomics, inclusion of therapy-related AMLs, and first and second-line therapeutic approaches. Second, in the response-to-therapy assessment, the authors did not include any MRD data, importantly impacting on AML outcomes. Additionally, the study design and results are complex and do not immediately impact on the reader as presented in the paper, especially regarding first-line and second-line treatment effects on outcomes.

Major revisions:

  1. Simple summary and Abstract do not adequately present the main manuscript and requires extensive revisions:

- simple summary, lines 51-52 contrast with discussion, regarding the efficacy of FLT3-inhibitor therapy. Additionally, no data on FLT3-inhibitors are reported in the manuscript results section.

Authors’ response: The simple summary has been revised to remove mention of FLT3-inhibitor therapy.

-simple summary, line 53: the manuscript focuses on FLT3 positive R/R AMLs: please edit the text

Authors’ response: This sentence has been revised to specify that patients in the study had relapsed/refractory AML with an FMS-like tyrosine kinase 3 mutation.

- simple summary, lines 54-59: please clarify whether salvage therapy refers to second-line treatment only and does not include first-line treatment data.

Authors’ response: This sentence has been revised to clarify that patients received salvage therapy refers to second-line treatment.

- simple summary, lines 59-60: the main manuscript does not support this conclusion

Authors’ response: The concluding statement of the simple summary has been revised to be consistent with the results of the main manuscript.

- the abstract should include Introduction, Material and Methods, and Conclusions section/information, beyond the results. Please integrate it

Authors’ response: Please note that the journal’s Instructions for Authors states “The abstract should be a single paragraph and should follow the style of structured abstracts, but without headings”. Therefore, we have not included the headings, but have added a concluding statement after the results.

-abstract, line 65: please specify that active salvage therapy includes only intensive and non-intensive therapy.

Authors’ response: This sentence has been revised to clarify that active salvage therapy includes intensive and non-intensive therapy, as suggested.

-abstract, lines 69-72: please clarify whether OS data refer to first-line or second-line therapy.

Authors’ response: This sentence has been revised to clarify that the OS data was in response to salvage (i.e., second-line) therapy.

2) Materials and Methods section:

- inclusion and exclusion criteria are complex and reported unclearly. A Table summarizing them may improve the manuscript.

Authors’ response: A table summarizing the inclusion and exclusion criteria has been added to Section 2.2 of the manuscript. To avoid repetition, we have removed some of the inclusion and exclusion criteria from this section, as they now appear in the table (Table 1).

- line 121: the authors should clarify why they included secondary AMLs together with de novo AMLs, considering differences in outcomes among them. Survival analysis could be performed separately for these two groups.

Authors’ response: Although differences between de novo and secondary AML are commonly described at diagnosis, there is a lack of information regarding these differences in patients with R/R FLT3-ITD-mutated AML. Therefore, we have added description of overall survival analysis in patients with de novo AML versus secondary AML (Section 3.4, page 9 and Figure 2d).

- lines 131-132: the authors should detail in the supplementary section the adopted methods for FLT3-ITD detection, especially considering that the analysis was performed by different laboratories without centralization to a referral lab.

Authors’ response: We thank the reviewer for this suggestion. We have added description of the methods of FLT3-ITD detection to the supplementary materials. Furthermore, we have provided details of the quality control system for the PETHEMA registry, whereby a biannual/annual exchange of samples was done between laboratories to ensure the reproducibility of genetic testing results.

- line 132: the authors should explain why they included patients without FLT3-ITD assessment at first diagnosis, as well as those patients showing FLT3-ITD only at relapse (Table 1). This could affect the study results.

Authors’ response: This was a real-world study with the goal of shedding light on real-life outcomes in patients with R/R FLT3-ITD-mutated AML, independent of their mutation status at diagnosis. Therefore, patients with FLT3-ITD mutations at relapse were also included.

- line 133: please define active first-line therapy. Considering therapy data complexity, the authors should name first-line (front-line) and second-line (salvage) therapy uniformly all over the manuscript.

Authors’ response: Details of active first-line therapy are provided in the text in Section 3.1 of the manuscript and in the Supplementary Materials (Supplemental Figure S1). The terminology has been revised throughout the manuscript to ‘first-line therapy’ (rather than ‘front-line therapy’) and ‘salvage therapy’ (rather than ‘second-line therapy’), as suggested.

-line 135: please clarify why second-generation FLT3-inhibitors were excluded and whether first-generation FLT3-inhibitor were included in the study

Authors’ response: Patients participating in clinical trials of FLT3 inhibitors as first-line treatment were included in intensive or non-intensive therapy groups according to the corresponding planned treatment. This information has not been detailed as most studies had a double-blind design and we could not confirm whether or not patients received any FLT3 inhibitor as first-line treatment. We have modified the manuscript to clarify this point (Section 2.4, page 4).

-lines 138-150: it would be useful to include response to therapy evaluation by MFC-MRD, to better understanding subsequent outcomes data.

Authors’ response: Determination of MRD activity at R/R AML diagnosis is not the standard of care; therefore, no treatment response data are available with regard to this topic.

- lines 142-146: the authors should clarify why they did not classify AMLs according to the 2016 Revision of WHO classification.

Authors’ response: The patients in our retrospective study were diagnosed with AML between 1998 and 2018. Consequently, as many patients were diagnosed prior to 2016, there are insufficient data to classify all patients under 2016 WHO criteria.

-line 144: please specify which genetic aberrations are included in each cytogenetic category

Authors’ response: As described in Table 2, cytogenetic risk was classified according to Medical Research Council criteria. We have also added some text in the Methods (Section 2.3, page 4) and a reference (Grimwade D, et al. Blood 2010, 116, 354-365) to further clarify the criteria for cytogenetic risk classification.

-lines 152-165: please add some information on SCT (first-line or second line treatment, or both?).

Authors’ response: As detailed in the text, SCT is related to second-line therapy. However, we have revised the Section 3.3 subheading to clarify this point (page 9). We have also added information regarding previous SCT to Table 3 (page 8).

- line168: please define partial remission

Authors’ response: According to the modified International Working Group criteria (Cheson BD, et al. J Clin Oncol 2003, 21, 4642-4649; now reference #25), partial remission “requires all of the hematologic values for a CR but with a decrease of at least 50% in the percentage of blasts to 5% to 25% in the bone marrow aspirate.” We have added this description to Section 2.5 (page 4) of the manuscript.

- line 169: please clarify whether induction death includes death for any cause or only disease-related death

Authors’ response: Induction death includes patients who died before undergoing response assessment. Based on the reviewer’s comment, we have revised the text in Section 2.5 (page 4) to include definition of induction death as follows: “Secondary endpoints were morphologic CR, CRi, and partial remission (PR; defined according to modified International Working Group criteria [25], induction death (defined as patients who died after starting salvage therapy but before being assessed), event-free survival (EFS), and frequency of subsequent allo-SCT.

5) Results:

- line 228: it would be interesting to have data on the time interval between first- and R/R diagnosis

Authors’ response: This information is already included in Section 3.2 (page 7) of the manuscript, where the mean time between initial AML diagnosis and subsequent diagnosis of R/R disease was shorter in the supportive care only group compared with intensive or non-intensive treatment groups, although this difference was not statistically significant. Moreover, we have provided a definition for the ‘Time to R/R’ variable as a footnote below Table 3.

- lines 267-268: please report the reasons for missing follow-up data in 44 patients

Authors’ response: Follow-up data were not missing for 44 patients, but response data were only available in 280 patients. Therefore, we have corrected this sentence on page 9 as follows: “Among the 324 patients who received active salvage therapy, response to treatment data were available in 280 patients (86%; Table 4).

-lines 267-277: please move to survival data section

Authors’ response: The sentence describing the patients who died during salvage therapy before the first treatment assessment has been shifted to the survival analysis section.

- survival analysis: it would be interesting to perform a multivariable analysis to show what effectively impacts on outcomes, considering the complexity of the study-cohort.

Authors’ response: We thank the reviewer for this suggestion. We have now conducted a multivariate analysis of overall survival and have added it as Section 3.4.1 (pages 11–12) to the manuscript. We have also added description of the methods of this multivariate analysis to Section 2.6 (page 4).

4) Table 1:

- the age range 17-86 contrasts with the inclusion criterium age ≥ 18 years. Please clarify this discrepancy

Authors’ response: We thank the reviewer for highlighting this discrepancy, which has now been corrected.

- it would be interesting to analyze young adults (18-25 years old) as a separate category, considering the biological differences of AMLs in this age group compared to older patients.

Authors’ response: Although this analysis would be interesting, we consider that this is out of the scope of the current manuscript and would undoubtedly entail increasing the overall length of the article, which is contrary to what was suggested by the previous reviewer.

- the authors may add a raw reporting front-line therapy data, including any SCT, if performed. The same could be done in Table 2

Authors’ response: We have added information regarding previous SCT to Table 3, as suggested.

5) Figure 2c: the 2nd Induction category shown in figure 2c is not cited in the manuscript. Please clarify it.

Authors’ response: The definition of ‘second induction’ has been cited in the main text of the manuscript in section 2.5 and in section 3.1.

6) Figure 2b: Induction death category should be removed from the analysis, being not comparable with the other three categories and potentially hampering OS results.

Authors’ response: We kindly disagree with this reviewer’s comment. ‘Induction death’ should be included in this figure as all patients received salvage treatment. Probably the confusion is due to a mistake in the figure legend, which we have now corrected to “(b) response to salvage treatment”.

7) Supplementary: it would be interesting to include a table matching first-line and second line treatments

Authors’ response: Although it would be interesting to include an analysis of first- and second-line treatment, we believe that this information would add complexity to the manuscript and make it difficult to understand, as this analysis would entail comparison between two groups that may include the same patients. Moreover, it would also further increase the overall length of the article, which is contrary to what was suggested by the previous reviewer.

Minor revisions:

1) Simple summary:

- line 55: 48% cannot be reported in digits at the beginning of a sentence. Please edit it.

Authors’ response: This sentence has been revised so it does not start with ‘48%’.

2) Materials and Methods:

-line 121: please define secondary AML, and then remove its definition from the Results section (line 212)

Authors’ response: Secondary AML is now defined in section 2.2 and removed from the results (section 3.1).

- lines 124, 126, 128, 141, 142, 147, 149, 162, 173: please substitute the i.e. with a complete list of all the definitions/variables pertinent to each specific brackets.

Authors’ response: The abbreviation ‘i.e.’ has been removed from these places in the manuscript.

-line 160: please detail LDAC therapy

Authors’ response: LDAC is defined as ‘low-dose AraC’ in section 2.2.

- lines 163-164: please remove Supplementary Table 1 referral from the Material and Methods section, because it belongs to the Results section.

Authors’ response: The sentence describing Supplemental Table S1 has been shifted to the start of section 3.2.

3) Results:

- line 217: please change allelic ratio to FLT3-ITD allelic ratio

Authors’ response: This text in section 3.1 (paragraph 2) has been revised.

-lines 278-285: the number of patients undergoing SCT after CR/CRi is reported twice. Please edit it.

Authors’ response: The text has been revised in section 3.3 to remove the repetition.

4) Table 1: age < 60 years, total, %: please change 501 to 51

Authors’ response: The number of patients aged <60 years has been corrected to 51 in the table.

5) Figures 2 and 3: please add p values

Authors’ response: The p-values for these survival analyses, which are describe in the text in Section 3.4 (pages 10 and 11), have been added to Figures 2 and 3, as requested.

Reviewer 3 Report

This retrospective study investigated outcomes of 404 patients with relapse/refractory (R/R) FLT3-internal tandem duplication acute myeloid leukemia (AML) enrolled in the PETHEMA registry, pre-approval of tyrosine kinase inhibitors. This real-world study showed that Median OS and EFS were very poor for both non-intensively and intensively treated patients, with best long-term outcomes obtained in patients achieving CR/CRi and subsequently receiving allo- SCT.

Although this real-world study in the pre-FLT3-inhibitors era is important to help inform the design of future clinical trials in this setting, there are several concerns.

My major concern is a lack of novelty. Multivariate analysis should be performed to draw definitive conclusions. What was the factor that affected OS other than allo-SCT?  

Author Response

#Reviewer 3

Comments and Suggestions for Authors

This retrospective study investigated outcomes of 404 patients with relapse/refractory (R/R) FLT3-internal tandem duplication acute myeloid leukemia (AML) enrolled in the PETHEMA registry, pre-approval of tyrosine kinase inhibitors. This real-world study showed that Median OS and EFS were very poor for both non-intensively and intensively treated patients, with best long-term outcomes obtained in patients achieving CR/CRi and subsequently receiving allo- SCT.

Although this real-world study in the pre-FLT3-inhibitors era is important to help inform the design of future clinical trials in this setting, there are several concerns.

My major concern is a lack of novelty.

Authors’ response: Although we understand the reviewer's concern, we kindly disagree with this comment as very few studies have addressed real-life experience in this specific cohort of patients. Moreover, to our knowledge, this is the largest real-world study focusing specifically on outcomes in patients with FLT3-ITD mutation-positive R/R AML.

Multivariate analysis should be performed to draw definitive conclusions.

Authors’ response: We thank the reviewer for this suggestion. As stated above, we have now conducted a multivariate analysis of overall survival and have added it as Section 3.4.1 (pages 11–12) to the manuscript.

What was the factor that affected OS other than allo-SCT?  

Authors’ response: As shown in the multivariate analysis, lack of IDH mutation and receiving a second identical induction cycle had a positive impact on OS, while not receiving active salvage therapy, being aged 60–70 years, and developed resistance after salvage therapy were independent predictors of shortened OS. This has been added to Section 3.4.1 (pages 11–12).

Round 2

Reviewer 1 Report

The authors have adequately addressed the issues that had been raised. No further comments from my side.

Reviewer 3 Report

The authors responded to my concerns.